# First whole-genome detection of dengue virus in urban *Aedes aegypti* from Southern Brazil

**Amanda Cupertino de Freitas**[1,2☉], **Sara Cândida Ferreira dos Santos**[1,3,4☉],
**Luiz Marcelo R. Tomé**[4], **Vagner Fonseca**[5], **Talita Émile Ribeiro Adelino**[1,4],
**Natália R. Guimarães**[1,4], **Felipe C. M. Iani**[4], **Keldenn Moreno**[1,3,4],
**Bruna Regina Diniz Souza**[2], **Victor Augusto Isidoro Maia**[2], **Getúlio Dornelles Souza**[6],
**Ellen Caroline Nobre Santos**[1], **Lívia Victória Rodrigues Baldon**[1], **Rafaela Luiza Moreira**[1],
**Maria Eduarda Calazans Rodrigues**[1], **Isaque João da Silva de Faria**[7],
**João Paulo Pereira de Almeida**[7], **Luiz Carlos Junior Alcantara**[1], **Marta Giovanetti**[8,9☉‡*],
**Álvaro Gil Araujo Ferreira**[1☉‡]

**1** Instituto René Rachou, Fundação Oswaldo Cruz, Belo Horizonte, Brazil, **2** Ecovec, Belo Horizonte, Brazil, **3** Programa Interunidades de Pós-graduação em Bioinformática, Universidade Federal de Minas Gerais, Belo Horizonte, Brazil, **4** Universidade Federal do Rio de Janeiro, Rio de Janeiro, Brazil, **5** Departamento de Ciências Exatas e da Terra, Universidade do Estado da Bahia, Salvador, Brazil, **6** Vigilância de Roedores e Vetores da Secretaria Municipal de Saúde (CGVS/SMS), Porto Alegre, Brazil, **7** Departamento de Bioquímica e Imunologia, Instituto de Ciências Biológicas, Universidade Federal de Minas Gerais, Belo Horizonte, Brazil, **8** Department of Science and Bio-Technology, Universita Campus Bio-Medico di Roma, Roma, Italy, **9** Oswaldo Cruz Institute, Oswaldo Cruz Foundation, Rio de Janeiro, Brazil

☉ These authors are contributed equally to this work.
‡ MG and ÁGAF are co-senior authors.
* giovanetti.marta@gmail.com

## Abstract

Dengue virus (DENV) is a major global health threat whose expansion into temperate regions has been facilitated by climate change and vector adaptation. Despite recurrent epidemics in Brazil, genomic surveillance in mosquito vectors remains limited, particularly in southern regions, constraining our understanding of local transmission dynamics and viral evolution. Here, we aimed to investigate dengue virus circulation, serotype co-circulation, and genomic signals of viral adaptation through vector-based genomic surveillance in southern Brazil. Using this approach, we provide genomic evidence of dengue virus circulation in southern Brazil during 2023. Whole-genome sequencing revealed active circulation of both DENV-1 (Genotype V) and DENV-2 (Genotype II), with the detection of mosquito pools harboring both serotypes, indicating their simultaneous circulation and raising concerns about increased transmission complexity and sequential infection risk in humans. Phylogenetic analyses supported sustained local transmission alongside multiple viral introductions. Moreover, recurrent mutations, particularly in non-structural proteins (NS1, NS2A, and NS5), suggest ongoing viral adaptation. These findings represent the first vector-derived genomic data for dengue in southern Brazil and highlight the critical role of mosquito-based genomic surveillance in detecting co-circulating serotypes, monitoring viral evolution, and strengthening preparedness in emerging transmission settings.

**Data availability statement:** Newly generated data have been submitted on NCBI under accession numbers: PX118979-PX119007.

**Funding:** This work was supported by the Novo Nordisk Foundation (NNF24OC0094346 to MG). The funders had no role in study design, data collection and analysis, decision to publish, or preparation of the manuscript. No authors received a salary from the funder for this work.

**Competing interests:** The authors have declared that no competing interests exist.

## Author summary

Dengue virus (DENV) continues to expand into new regions, driven by climate change and mosquito adaptation. In Brazil, dengue epidemics are recurrent, but genomic surveillance of mosquitoes is still scarce, particularly in the southern states. To fill this gap, we implemented an urban mosquito-trapping strategy designed for low-density and peri-domestic environments in Porto Alegre, Rio Grande do Sul. From April to July 2023, we collected 4,768 *Aedes aegypti* samples across 16 neighborhoods, generating 2,022 pools. Among these, 41 pools tested positive for DENV, of which 33 pools exhibited RNA integrity suitable for sequencing. The analysis revealed circulation of DENV-1 (Genotype V) and DENV-2 (Genotype II), with two pools showing co-circulation of both serotypes, raising concern for sequential infections in humans. Phylogenetic analysis revealed sustained local transmission combined with multiple virus introductions, and mutations in viral proteins suggested ongoing viral adaptation. Our findings provide the first mosquito-based genomic data for dengue in southern Brazil and highlight how mosquito genomics can strengthen early outbreak detection, serotype monitoring, and epidemic preparedness.

## Introduction

Dengue fever is a mosquito-borne disease caused by *Orthoflavivirus denguei* (DENV), a single-stranded, positive-sense RNA virus in the *Flaviviridae* family. The viral genome is approximately 10.7 kb in length and encodes three structural proteins (C, prM/M, and E) and seven nonstructural proteins (NS1–NS5). It comprises four serotypes (DENV-1 to DENV-4), each with multiple genotypes. Transmission occurs via infected *Aedes* mosquitoes—primarily *Aedes aegypti* and *Aedes albopictus*—and can lead to a wide range of clinical outcomes, from mild febrile illness to severe disease and death [1,2]. Environmental and anthropogenic factors such as climate change, urbanization, deforestation, and human mobility have facilitated the global expansion of dengue [1,2]. Temperature, in particular, plays a central role in viral transmission and vector competence. Optimal transmission occurs between 32°C and 33°C, although infection can persist at lower thresholds [1,2]. Elevated temperatures can enhance mosquito activity, modify morphology, and improve viral replication by affecting structural proteins such as the envelope (E) protein. Studies have also shown that increasing temperatures improve *Aedes albopictus* ability to transmit DENV-2 by facilitating viral dissemination from the midgut to the salivary glands [1,2]. In recent years, vector-based genomic surveillance has emerged as a valuable approach to directly investigate dengue virus circulation and evolution within mosquito populations. Studies conducted in dengue-endemic regions of Southeast Asia and Central America have demonstrated that mosquito-derived genomic data can reveal cryptic transmission, co-circulation of serotypes, and early signals of viral diversification that are not always captured through clinical surveillance alone (6–9).

However, such approaches remain unevenly implemented globally and are largely absent from temperate or transitional regions experiencing recent dengue emergence. DENV was first reported in Brazil during epidemics in 1846–1848 and 1851–1853, with additional outbreaks in 1916 and 1923 [3,4]. After re-emerging in 1982 in Roraima, major outbreaks followed, notably in Rio de Janeiro in 1986 with the introduction of DENV-1 and the first detection of *Aedes albopictus* in the country [3]. Subsequent introductions included DENV-2 in 1990, DENV-3 in 2000, and DENV-4 in 2012–2013. In 2019, Brazil reported over 1.5 million cases and 782 deaths [4]. While the Southeast, Northeast, and Central-West regions are hyperendemic, the South—particularly Rio Grande do Sul (RS)—was historically less affected due to its temperate climate. However, climate change and vector adaptation have driven dengue expansion in the region [5]. *Aedes aegypti* was first detected in RS in 1995, and the first autochthonous case was reported in 2007. The virus subsequently spread to 453 of 497 municipalities by 2022 [5]. From 2021–2022, RS recorded 101,481 cases and 121 deaths [5]. In 2023, 38,176 cases and 54 deaths were confirmed [6,7]. By 2024, RS experienced its largest recorded outbreak, with 207,465 confirmed cases, a 9.2% case-fatality rate, and *Aedes aegypti* detected in 94.9% of municipalities [6,7]. Despite this rapid epidemiological expansion, genomic surveillance in southern Brazil remains largely restricted to human cases, with a marked scarcity of vector-derived genomic data. This lack of mosquito-based genomic information limits our ability to resolve local serotype circulation, transmission dynamics, and viral evolutionary processes in a region transitioning from low to sustained dengue transmission. The aim of this study was therefore to characterize circulating dengue virus serotypes and investigate genomic signatures of viral transmission and adaptation in southern Brazil using vector-based genomic surveillance, given its intrinsic link to viral transmission cycles and evolution. To achieve this, we applied an urban mosquito-trapping strategy optimized for low-density and peri-domestic environments.

## Results and methods

Between April and July 2023, we collected 4,768 *Aedes aegypti* mosquitoes across 16 neighborhoods of Porto Alegre, spanning five macro-regions defined by the city's epidemiological planning framework: East Zone (Partenon, Jardim Botânico, Santa Rosa de Lima), North Zone (Sarandi, São Sebastião, Passo das Pedras), South Zone (Tristeza, Nonoai), Central-East Zone (Santa Tereza), and Northeast Zone (Vila João Pessoa, Vila São José, Mário Quintana, Bom Jesus, Costa e Silva, Vila Ipiranga) (Fig 1a). Our sampling strategy prioritized ecological and socio-spatial diversity and was designed to capture urban areas typically underrepresented in routine entomological surveillance. Adult *Aedes aegypti* were collected using Mosquitrap devices (Ecovec, Brazil) deployed at 250-m intervals across urban areas and checked weekly. Mosquitoes from each trap were pooled (males and females combined [8]) into tubes containing 250 µL of lysis buffer with ten 0.1-mm zirconium beads for mechanical homogenization, yielding a total of 2,022 pools according to collection site and date (S1 Table). Total RNA was extracted from homogenized samples using the BioGene Viral DNA/RNA Extraction Kit (Bioclin, Brazil), following the manufacturer's silica-column purification protocol. RNA was eluted in 20 µL of RNase-free water and screened by RT-qPCR for dengue virus (DENV) serotypes 1–4 to identify candidates for genome sequencing [9–11]. Following screening, 41 pools tested positive for DENV, and 33 exhibited RNA integrity suitable for sequencing (S1 Table). High-throughput whole-genome sequencing was then performed to assess DENV presence, determine serotype distribution, and characterize viral genetic diversity within local mosquito populations.

Complementary DNA (cDNA) was synthesized using SuperScript IV Reverse Transcriptase (Thermo Fisher Scientific, Waltham, MA, USA), followed by amplification multiplex PCR with Q5 High-Fidelity Hot-Start DNA Polymerase (New England Biolabs, Ipswich, MA, USA) and serotype-specific primer sets designed by the CADDE project (https://www.caddecentre.org/) for sequencing the complete genomes of DENV-1 and DENV-2 [4,9,10]. Amplicons were purified using 1×AMPure XP beads (Beckman Coulter, Brea, CA, USA) and quantified with a Qubit 3.0 fluorometer and Qubit dsDNA HS Assay Kit (Thermo Fisher Scientific) [9,11]. Sequencing libraries were prepared using the Illumina COVIDSeq Test, following the manufacturer's protocol, and sequenced on the MiSeq platform with a 300-cycle MiSeq Reagent Kit v2. Raw reads were trimmed using Trimmomatic 0.39 and mapped to the reference genome (GISAID: EPI_ISL_17983078 and

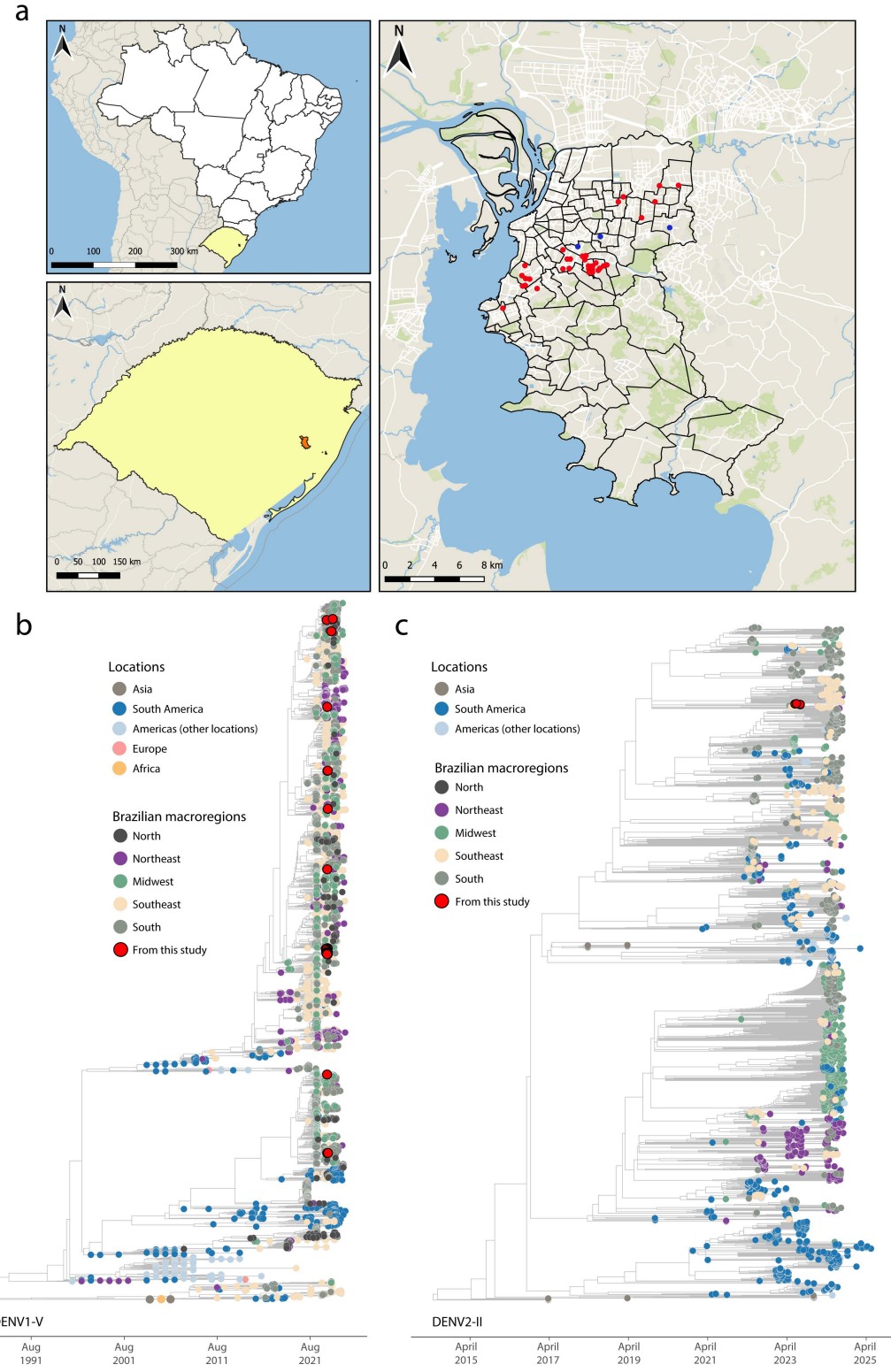

**Fig 1. Phylogenetic and spatial analysis of DENV from Aedes mosquitoes in Porto Alegre, Brazil. A)** Maps showing the spatial area under investigation. Red dots indicate mosquito collection sites. The base layer (administrative boundaries) was obtained using the geobr R package, which provides official Brazilian territorial boundary datasets ("Malhas Territoriais") from the Brazilian Institute of Geography and Statistics (IBGE).; **B)** Maximum

likelihood (ML) phylogenetic tree of 29 complete DENV-1 genome sequences generated in this study, analyzed alongside 5,635 reference sequences from GISAID and GenBank. Colors indicate sampling locations by global region and Brazilian macroregion; **C)** ML phylogenetic tree of 4 complete DENV-2 genome sequences generated in this study, analyzed with 1,830 reference sequences from GISAID and GenBank. Colors indicate sampling locations. For the base layer (administrative boundaries), we used the geobr R package, which downloads official Brazilian territorial boundary datasets ("Malhas territoriais") from IBGE. Base layer source: geobr package (CRAN/ project page) https://cran.rproject.org/web/packages/geobr/geobr.pdf and IBGE Malhas Territoriais https://www.ibge.gov.br/geociencias/organizacao-do-territorio/malhas-territoriais.html.

EPI_ISL_19609373) with Minimap2 2.30-r1287 Subsequent processing included sorting and indexing with Samtools 1.22, indel correction with Pilon 1.24. Low-depth positions (<5×) were masked as 'N', and only genomes with >60% coverage were retained for phylogenetic analysis [12]. Sequences were aligned using MAFFT v7.505 and manually inspected in AliView v1.30 to ensure alignment quality [4]. A maximum likelihood tree was inferred using IQ-TREE3 with the GTR+G4+F model. Temporal signal was assessed with TempEst v41, and outlier sequences showing clear deviations from the expected linear relationship between genetic divergence and sampling time were removed to improve dating accuracy. The final time-scaled phylogeny was reconstructed with TreeTime using a constant rate model, and a discrete trait analysis was performed to infer viral movements across Brazilian regions and globally [4,12].

Of the 41 mosquito pools positive in the screening, each represented the total number of mosquitoes collected from a single trap over a 7-day period, ranging from 1 to 12 individuals, with a total of 149 mosquitoes. Thirty-three yielded sufficient RNA for whole-genome sequencing and tested positive for DENV, 29 for DENV-1 and 4 for DENV-2. This marks the first genomic detection of dengue virus in field-caught mosquitoes from southern Brazil. Two pools (ID_LAB: 4542341, 4539959) showed co-detection of both serotypes, indicating simultaneous circulation of DENV-1 and DENV-2 within the same vector population. While this does not confirm co-infection at the individual mosquito level, it has epidemiological relevance due to the risk of sequential infections in humans. Positive samples showed Ct values ranging from 19.79 to 37, with most below 25, reflecting high viral loads. Sequencing depth ranged from 54× to 2,081×, with read counts between 8,090 and 222,964 for each sequenced sample. Genome coverage ranged from 61.6% to 93.0%, with over 75% of samples achieving >85% coverage, supporting reliable genotyping and phylogenetic analysis (S1 Table).

All DENV-1 genomes were classified as Genotype V (lineages E.1 and D.1.1) and all DENV-2 genomes as Genotype II (lineage F.1.1.2), both previously reported in Brazil [13,14]. DENV-1 was widely distributed across Porto Alegre's five macro-regions, while DENV-2 was geographically limited to the East and Northeast zones (Fig 1a). Phylogenetic analysis revealed that DENV-1 sequences from this study clustered into four distinct clades (Fig 1b). Most grouped with sequences from southern Brazil, consistent with ongoing local transmission, while others clustered with viruses from different regions, suggesting multiple introductions and regional viral exchange. DENV-2 sequences were nested among those previously sampled from the South, preceding the 2024 outbreak across southern and southeastern Brazil (Fig 1c).

Additionally, all generated genomes were screened for mutations and annotated using SnpEff to predict potential functional impacts. Most mutations in DENV-1 and DENV-2 genomes were synonymous, with low predicted impact as they do not alter the amino acid sequence of viral proteins. Missense mutations, which can have a moderate impact through amino acid substitutions, were identified in both serotypes. In DENV-1, these were restricted to non-structural proteins—NS1 (n=4), NS2A (n=3), NS3 (n=1), NS4B (n=1), and NS5 (n=1)—with NS1 and NS2A most frequently affected (S2 Table). In DENV-2, missense mutations occurred in both structural (E, n=3) and non-structural proteins—NS1 (n=1), NS2A (n=1), NS3 (n=2), NS4B (n=1), and NS5 (n=5)—with the highest frequency observed in NS5 (S2 Table). Many of these mutations, however, were not unique to our dataset; they have been reported in Brazilian genomes from both current and past epidemics, suggesting a long-standing trend toward viral adaptation. Although their precise functional implications remain to be determined, the recurrent occurrence of changes in proteins involved in replication and immune evasion may reflect selective pressures acting on dengue viruses in Brazil—a hypothesis that warrants further investigation [15].

## Discussion

Genomic surveillance of mosquito-borne arboviruses remains substantially less developed than surveillance based on human cases, creating a persistent blind spot in our understanding of viral circulation, evolution, and persistence within vector populations. This imbalance is particularly relevant given that mosquitoes constitute the ecological interface where viral maintenance, diversification, and onward transmission occur [1,2,5–8,13–15]. Technical constraints—most notably low viral loads and heterogeneous infection rates in field-collected mosquitoes—have historically limited the routine integration of vector genomics into surveillance frameworks, leaving key transmission processes largely inferred rather than directly observed [1,3,13].

Within this context, our findings illustrate how vector-based genomic surveillance can refine the interpretation of dengue transmission dynamics in regions undergoing epidemiological transition. Southern Brazil has traditionally been considered peripheral to sustained dengue transmission; however, recent climatic and ecological changes have altered vector suitability and epidemic potential [2–5]. The genomic evidence presented here indicates that dengue transmission in this region is not sporadic or incidental but instead embedded within broader national transmission networks. The observed phylogenetic patterns are consistent with continuous viral connectivity across Brazilian regions rather than isolated spillover events, reinforcing the notion that geographic and climatic boundaries provide diminishing barriers to arbovirus spread [1–5].

Beyond documenting viral presence, vector-derived genomic data provide insights into viral evolutionary processes that are difficult to capture through human case surveillance alone [2,6,8]. In this study, the majority of detected mutations in DENV-1 and DENV-2 genomes were synonymous, consistent with strong purifying selection acting on dengue virus populations. Missense substitutions were less frequent and primarily involved non-structural proteins (NS) associated with viral replication and host immune interactions. Notably, many of these amino acid changes have been previously reported in Brazilian genomes from both recent and historical outbreaks, suggesting persistent selective pressures rather than the emergence of novel, lineage-specific mutations [15]. Although the functional consequences of these substitutions remain to be determined, their recurrent detection underscores the potential value of vector-based genomics as an early indicator of viral diversification with epidemiological relevance.

Our results highlight the strategic importance of incorporating vector-based genomics into dengue monitoring frameworks, particularly in regions where human case surveillance may lag true transmission dynamics. As dengue continues to expand into temperate and previously low-risk regions, integrating mosquito-derived genomic data will be essential for improving early detection, refining risk assessment, and informing adaptive public health and vector control strategies in an evolving climatic and epidemiological landscape [2–5].

## Limitations

This study has some limitations. The use of pooled mosquito samples limits the ability to infer individual infection rates and may reduce sensitivity for detecting low-prevalence infections. In addition, the cross-sectional design and restriction to a four-month sampling window preclude the assessment of seasonal patterns or temporal changes in dengue virus circulation. Finally, sequencing success was dependent on RNA integrity, resulting in fewer complete genomes than the total number of DENV-positive pools. Despite these limitations, our study provides important strengths. It represents the first vector-based whole-genome characterization of dengue virus in southern Brazil and demonstrates the feasibility and value of mosquito-based genomic surveillance in a region undergoing rapid epidemiological transition. The integration of entomological sampling with high-throughput sequencing enabled the direct detection of co-circulating serotypes and the identification of genomic signals relevant to transmission dynamics. Together, these data establish a robust foundation for future longitudinal, spatially expanded surveillance efforts and support the inclusion of vector genomics as a complementary component of dengue monitoring frameworks.

## Supporting information

**S1 Table. Metadata of the samples analyzed in this study, including information on the collections performed, PCR and sequencing results, and the accession numbers of the sequences deposited in GenBank.**
(XLSX)

**S2 Table. Information on mutations found according to the reference used, specifying the nucleotide substitution, effect, impact, amino acid change, the protein affected by the mutation, and the number of target genomes (from mosquitoes) in which these detections were grouped.**
(XLSX)

## Author contributions

**Conceptualization:** Victor Augusto Isidoro Maia, Marta Giovanetti, Álvaro Gil Araujo Ferreira.

**Data curation:** Amanda Cupertino de Freitas, Sara Cândida Ferreira dos Santos, Luiz Marcelo R. Tomé, Vagner Fonseca, Marta Giovanetti.

**Formal analysis:** Amanda Cupertino de Freitas, Sara Cândida Ferreira dos Santos, Luiz Marcelo R. Tomé, Vagner Fonseca, Talita Émile Ribeiro Adelino, Felipe C.M. Iani, Keldenn Moreno, Bruna Regina Diniz Souza, Victor Augusto Isidoro Maia, Getúlio Dornelles Souza, Lívia Victória Rodrigues Baldon, Rafaela Luiza Moreira, Isaque João da Silva de Faria, Marta Giovanetti.

**Funding acquisition:** Ellen Caroline Nobre Santos, Marta Giovanetti.

**Investigation:** Amanda Cupertino de Freitas, Sara Cândida Ferreira dos Santos, Luiz Marcelo R. Tomé, Vagner Fonseca, Talita Émile Ribeiro Adelino, Natália R. Guimarães, Felipe C.M. Iani, Keldenn Moreno, Bruna Regina Diniz Souza, Victor Augusto Isidoro Maia, Getúlio Dornelles Souza, Ellen Caroline Nobre Santos, Lívia Victória Rodrigues Baldon, Rafaela Luiza Moreira, Maria Eduarda Calazans Rodrigues, Isaque João da Silva de Faria, João Paulo Pereira de Almeida, Luiz Carlos Junior Alcantara, Marta Giovanetti, Álvaro Gil Araujo Ferreira.

**Methodology:** Talita Émile Ribeiro Adelino, Natália R. Guimarães, Felipe C.M. Iani, Bruna Regina Diniz Souza, Victor Augusto Isidoro Maia, Getúlio Dornelles Souza, Ellen Caroline Nobre Santos, Lívia Victória Rodrigues Baldon, Rafaela Luiza Moreira, Isaque João da Silva de Faria, João Paulo Pereira de Almeida, Luiz Carlos Junior Alcantara, Marta Giovanetti, Álvaro Gil Araujo Ferreira.

**Project administration:** Marta Giovanetti, Álvaro Gil Araujo Ferreira.

**Validation:** Marta Giovanetti.

**Visualization:** Marta Giovanetti.

**Writing – original draft:** Sara Cândida Ferreira dos Santos, Marta Giovanetti.

**Writing – review & editing:** Amanda Cupertino de Freitas, Sara Cândida Ferreira dos Santos, Luiz Marcelo R. Tomé, Vagner Fonseca, Talita Émile Ribeiro Adelino, Natália R. Guimarães, Felipe C.M. Iani, Keldenn Moreno, Bruna Regina Diniz Souza, Victor Augusto Isidoro Maia, Getúlio Dornelles Souza, Ellen Caroline Nobre Santos, Lívia Victória Rodrigues Baldon, Rafaela Luiza Moreira, Maria Eduarda Calazans Rodrigues, Isaque João da Silva de Faria, João Paulo Pereira de Almeida, Luiz Carlos Junior Alcantara, Marta Giovanetti, Álvaro Gil Araujo Ferreira.

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
