## [Decision Letter · Decision Letter 0]

14 Nov 2025

Short report: First whole-genome evidence of dengue virus in field-caught mosquitoes from southern Brazil

Dear Dr. Giovanetti,

Thank you for submitting your manuscript to PLOS Neglected Tropical Diseases. After careful consideration, we feel that it has merit but does not fully meet PLOS Neglected Tropical Diseases's publication criteria as it currently stands. Therefore, we invite you to submit a revised version of the manuscript that addresses the points raised during the review process.

Please submit your revised manuscript within by Jan 13 2026 11:59PM. If you will need more time than this to complete your revisions, please reply to this message or contact the journal office at plosntds@plos.org. Please include the following items when submitting your revised manuscript:

We look forward to receiving your revised manuscript.

Kind regards,

Kinley Wangdi, PhD

Academic Editor

Michael Holbrook

Section Editor

Shaden Kamhawi

co-Editor-in-Chief

Paul Brindley

co-Editor-in-Chief

**Additional Editor Comments:**

Abstract

The aim should be made clear.

Introduction

The aim is not clear; it should be made clear. Is the aim mosquito-trapping or quantify the circulating dengue virus serotypes.

The study

1. The process of sampling is very detailed, summarize the process.

2. Limitation of study is needed.

**Journal Requirements:**

At this stage, the following Authors/Authors require contributions: Maria Eduarda Calazans Rodrigues, and Isaque João da Silva de Faria. Please ensure that the full contributions of each author are acknowledged in the "Add/Edit/Remove Authors" section of our submission form.

- ® on page: 4.

4) We notice that your supplementary figures are uploaded with the file type 'Table'. Please amend the file type to 'Supporting Information'. Please ensure that each Supporting Information file has a legend listed in the manuscript after the references list.

Potential Copyright Issues:

- Figure 1. Please (a) provide a direct link to the base layer of the map (i.e., the country or region border shape) and ensure this is also included in the figure legend; and (b) provide a link to the terms of use / license information for the base layer image or shapefile. We cannot publish proprietary or copyrighted maps (e.g. Google Maps, Mapquest) and the terms of use for your map base layer must be compatible with our CC BY 4.0 license.

**Reviewers' Comments:**

Reviewer's Responses to Questions

**Key Review Criteria Required for Acceptance?**

**Methods**

-Are the objectives of the study clearly articulated with a clear testable hypothesis stated?

-Is the study design appropriate to address the stated objectives?

-Is the population clearly described and appropriate for the hypothesis being tested?

-Is the sample size sufficient to ensure adequate power to address the hypothesis being tested?

-Were correct statistical analysis used to support conclusions?

-Are there concerns about ethical or regulatory requirements being met?

Reviewer #1: (No Response)

Reviewer #2: The manuscript demonstrated the whole-genome evidence of dengue virus in field-caught mosquitoes from southern Brazil. The method is precise; however, some issues need to be clarified.

Reviewer #3: (No Response)

**Results**

-Does the analysis presented match the analysis plan?

-Are the results clearly and completely presented?

-Are the figures (Tables, Images) of sufficient quality for clarity?

Reviewer #1: (No Response)

Reviewer #2: The results presented is not clear and insufficient.

The figure 1, is not informative and very busy to comprehend. The authors should reconstruct the phylogenetic tree with the concise needed information (fewer reference sequences from the Genbank) and outgroup. The authors should also describe each clade clearly for the readers.

Line 135 -140: Not clearly demonstrated in the Figure 1B-C, very difficult to follow.

Reviewer #3: (No Response)

**Conclusions**

-Are the conclusions supported by the data presented?

-Are the limitations of analysis clearly described?

-Do the authors discuss how these data can be helpful to advance our understanding of the topic under study?

-Is public health relevance addressed?

Reviewer #1: (No Response)

Reviewer #2: (No Response)

Reviewer #3: * The manuscript currently does not include a clearly defined Discussion section. Adding this section would help contextualize the findings and relate them to existing literature.

* The manuscript would also benefit from the inclusion of a brief statement addressing the limitations of the study. Highlighting these limitations will provide a more balanced interpretation of the results and strengthen the overall presentation.

**Editorial and Data Presentation Modifications?**

Reviewer #1: (No Response)

Reviewer #2: (No Response)

Reviewer #3: * In the Background section, it would be helpful to include additional general information about the virus, such as the approximate size of its genome.

* The abbreviation “NS” should be formally defined at first mention.

* The manuscript presents genomes sequenced from specimens collected in 2023 and shown on a phylogenetic tree spanning isolates from 1991 to 2023 (Figure 1B). However, the compressed time scale makes it appear that the 2023 genomes are closer to the August 2021 time point, which may cause confusion. It would be useful to clearly mark or label the 2023 point on the timeline.

* The formatting of the Supplemental Tables needs correction. Several tables contain truncated text, overlapping content across pages, and columns that are too narrow—causing numerical values to appear as “###.” Please adjust the layout to ensure all text and data are clearly visible.

* The manuscript aims to address an important knowledge gap regarding DENV strains circulating in underrepresented regions of Brazil. While the methods used for genomic sequencing appear generally adequate, the methodological details are interspersed with the results and discussion. This formatting choice makes it difficult to clearly and critically evaluate the methods. The authors are encouraged to present the Methods section separately to improve clarity and reproducibility.

* Line 117: The authors state that “outliers were removed to improve dating accuracy.” Please provide additional detail on the parameters or criteria used to identify and remove these outliers, as this information is essential for evaluating the validity of the temporal analysis.d

**Summary and General Comments**

Reviewer #1: (No Response)

Reviewer #2: The manuscript demonstrated the whole-genome evidence of dengue virus in field-caught mosquitoes from southern Brazil. The method is precise; however, some issues need to be clarified and more discussed as follows.

The authors should provide citations to their statements below:

Line 58-59 “Environmental and anthropogenic factors……….”

Line 75-76 “ In 2023, 38,176 cases………”

The authors mentioned they pooled both males and females into tubes. What is the essence of including males in the study?

How long did the authors preserved the samples without significant RNA degradation and what is the medium of preservation (solution?)?

Line 103: typo “for” “for” delete one.

Regarding the RNA extraction, it can be found that the genomic DNA, if not digested, can be present in the eluted RNA, resulting possible positive amplification of the gene of interest in tested specimens. Did the authors decontaminate the DNA before amplification?

The authors should provide the serotype-specific primers used in the study?

The pooling method described in the study is inappropriately described. Most samples in the tubes analyzed were individual (1) samples, yet the authors mentioned they are pools. Pool samples indicates combination of more than 1 sample together in a tube.

Also, the one (1) sample in a tubes, is it male or female?

From the Table_S1, the study has unequal sample pools from 2-15 pools per tube, and individual (1) sample tubes.

What is the total number of males and females’ flies included in the study?

The 2 pools the author mentioned that were positive for both DENV I and 2, how many samples in each pool? All males or females?

The authors claimed that they had 33 positive samples out of 41 pools for DENV: 29 for DENV-1 and 4 for DENV-2. How accurate is this high infection rate? Does the author understand the epidemiological implication of this claim. The chance of possible transmission is very high in the study areas. These results are discussable.

How did the authors avoid contaminations amongst sample analyzed?

Line 127: Were the CT values analyzed by the comparative threshold cycle method, and normalized to the endogenous/reference control?

The figure 1, is not informative and very busy to comprehend. The authors should reconstruct the phylogenetic tree with the concise needed information (fewer reference sequences from the Genbank) and outgroup. The authors should also describe each clade clearly for the readers.

Line 135 -140: Not clearly demonstrated in the Figure 1B-C, very difficult to follow.

Overall, the observed results needs more discussion.

Reviewer #3: * There could be substantial additional value to the findings reported in this manuscript if the authors extended their analysis. While they allude to the importance of understanding the impact of the variants identified in the sequenced DENV genomes, they do not perform the bioinformatic analyses necessary to address this knowledge gap. With the availability of advanced structural modeling tools (e.g., AlphaFold3), it would be feasible to assess the potential structural or functional implications of the identified variants using a reference DENV strain as a model template. Including such analyses would considerably strengthen the study’s impact and interpretive depth.

PLOS authors have the option to publish the peer review history of their article (what does this mean? ). If published, this will include your full peer review and any attached files.

**Do you want your identity to be public for this peer review?** For information about this choice, including consent withdrawal, please see our Privacy Policy .

Reviewer #1: **Yes:** Jian-Wei Shao

Reviewer #2: No

Reviewer #3: No

**Figure resubmission:**
---

## [Decision Letter · Decision Letter 1]

11 Feb 2026

Response to Reviewers
Revised Manuscript with Track Changes
Manuscript

Shaden Kamhawi

co-Editor-in-Chief

Paul Brindley

co-Editor-in-Chief

**Additional Editor Comments:**
**Journal Requirements:**

At this stage, the following Authors/Authors require contributions: Maria Eduarda Calazans Rodrigues. Please ensure that the full contributions of each author are acknowledged in the "Add/Edit/Remove Authors" section of our submission form.

3) Tables should not be uploaded as individual files. Please remove these files and include the Tables in your manuscript file as editable, cell-based objects. For more information about how to format tables, see our guidelines:

https://journals.plos.org/plosntds/s/tables

4) Please amend your detailed Financial Disclosure statement. This is published with the article. It must therefore be completed in full sentences and contain the exact wording you wish to be published.

**Reviewers' comments:**

**Key Review Criteria Required for Acceptance?**

**Methods**

-Are the objectives of the study clearly articulated with a clear testable hypothesis stated?

-Is the study design appropriate to address the stated objectives?

-Is the population clearly described and appropriate for the hypothesis being tested?

-Is the sample size sufficient to ensure adequate power to address the hypothesis being tested?

-Were correct statistical analysis used to support conclusions?

-Are there concerns about ethical or regulatory requirements being met?

Reviewer #1: The objectives of the study clearly articulated with a clear testable hypothesis stated

Reviewer #2: (No Response)

Reviewer #3: (No Response)

**Results**

-Does the analysis presented match the analysis plan?

-Are the results clearly and completely presented?

-Are the figures (Tables, Images) of sufficient quality for clarity?

Reviewer #1: (No Response)

Reviewer #2: (No Response)

Reviewer #3: (No Response)

**Conclusions**

-Are the conclusions supported by the data presented?

-Are the limitations of analysis clearly described?

-Do the authors discuss how these data can be helpful to advance our understanding of the topic under study?

-Is public health relevance addressed?

Reviewer #1: (No Response)

Reviewer #2: (No Response)

Reviewer #3: (No Response)

**Editorial and Data Presentation Modifications?**

Reviewer #1: Having reviewed the authors' responses to my previous concerns, I find that all issues have been satisfactorily addressed. I now recommend that the manuscript be accepted for publication.

Reviewer #2: (No Response)

Reviewer #3: (No Response)

**Summary and General Comments**

Reviewer #1: (No Response)

Reviewer #2: (No Response)

Reviewer #3: The manuscript has improved considerably with the current revision. The authors have done a commendable job expanding the discussion and acknowledging the study’s shortcomings through the addition of a limitations section. However, a few minor issues remain that should be addressed prior to acceptance.

On lines 155–157, the authors state that the distributions of DENV-1 and DENV-2 occur in different geographic locations, as shown in Figure 1A. However, all data points appear in the same color, preventing the reader from distinguishing between DENV-1 and DENV-2 by region as suggested in the text.

Additionally, many values in Supplemental Table 1 are still displayed as “####” in the CT values, total reads, and Deep columns. The authors indicated that this issue was corrected during revision, but it does not appear to have been fully resolved.

Finally, there are multiple instances throughout the manuscript of extra spaces and an occasional additional period. The manuscript would benefit from another careful proofreading to address these minor typographical issues and to ensure consistent use of abbreviations throughout.

PLOS authors have the option to publish the peer review history of their article (what does this mean? ). If published, this will include your full peer review and any attached files.

**Do you want your identity to be public for this peer review?** For information about this choice, including consent withdrawal, please see our Privacy Policy .

Reviewer #1: **Yes:** Jian-Wei Shao

Reviewer #2: No

Reviewer #3: No

**Figure resubmission:**

**Reproducibility:** To enhance the reproducibility of your results, we recommend that authors of applicable studies deposit laboratory protocols in protocols.io, where a protocol can be assigned its own identifier (DOI) such that it can be cited independently in the future. Additionally, PLOS ONE offers an option to publish peer-reviewed clinical study protocols. Read more information on sharing protocols at https://plos.org/protocols?utm_medium=editorial-email&utm_source=authorletters&utm_campaign=protocols

---

## [Editor Report · Decision Letter 2]

17 Feb 2026

Dear Dr. Giovanetti,

We are pleased to inform you that your manuscript 'First Whole-Genome Detection of Dengue Virus in Urban Aedes aegypti from Southern Brazil' has been provisionally accepted for publication in PLOS Neglected Tropical Diseases.

Best regards,

Kinley Wangdi, PhD

Academic Editor

Michael Holbrook

Section Editor

Shaden Kamhawi

co-Editor-in-Chief

Paul Brindley

co-Editor-in-Chief

---

## [Editor Report · Acceptance letter]

Dear Dr. Giovanetti,

We are delighted to inform you that your manuscript, "First Whole-Genome Detection of Dengue Virus in Urban Aedes aegypti from Southern Brazil," has been formally accepted for publication in PLOS Neglected Tropical Diseases.

Best regards,

Shaden Kamhawi

co-Editor-in-Chief

Paul Brindley

co-Editor-in-Chief
